# A wheat cysteine-rich receptor-like kinase confers broad-spectrum resistance against Septoria tritici blotch

Cyrille Saintenac [1]✉, Florence Cambon[1], Lamia Aouini [2,8], Els Verstappen[2],
Seyed Mahmoud Tabib Ghaffary[2,9], Théo Poucet[1,10,11], William Marande[3], Hélène Berges[3,12], Steven Xu[4],
Maëlle Jaouannet[5], Bruno Favery [5], Julien Alassimone[6], Andrea Sánchez-Vallet [6,13], Justin Faris[4],
Gert Kema [2,14], Oliver Robert[7] & Thierry Langin[1]

The poverty of disease resistance gene reservoirs limits the breeding of crops for durable resistance against evolutionary dynamic pathogens. *Zymoseptoria tritici* which causes Septoria tritici blotch (STB), represents one of the most genetically diverse and devastating wheat pathogens worldwide. No fully virulent *Z. tritici* isolates against synthetic wheats carrying the major resistant gene *Stb16q* have been identified. Here, we use comparative genomics, mutagenesis and complementation to identify *Stb16q*, which confers broad-spectrum resistance against *Z. tritici*. The *Stb16q* gene encodes a plasma membrane cysteine-rich receptor-like kinase that was recently introduced into cultivated wheat and which considerably slows penetration and intercellular growth of the pathogen.

[1] Université Clermont Auvergne, INRAE, GDEC, 63000 Clermont-Ferrand, France. [2] Wageningen University and Research (Wageningen Plant Research, Biointeractions and Plant Health), PO Box 16, 6700AA Wageningen, The Netherlands. [3] CNRGV (Centre National des Ressources Génomiques Végétales), INRAE, UPR 1258 Castanet-Tolosan, France. [4] United States Department of Agriculture-Agricultural Research Service, Cereal Crops Research Unit, Edward T. Schafer Agricultural Research Center, Fargo, ND 58102, USA. [5] INRAE, Université Côte d'Azur, CNRS, ISA, 06903 Sophia Antipolis, France. [6] Plant Pathology, Institute of Integrative Biology, ETH Zürich, 8092 Zürich, Switzerland. [7] Florimond-Desprez, 3 rue Florimond-Desprez, BP 41, 59242 Cappelle-en-Pevele, France. [8] Present address: Department of Agronomy, Purdue University, West Lafayette, IN 47907, USA. [9] Present address: Seed and Plant Improvement Research Department, Safiabad Agricultural and Natural Resources Research and Education Center, AREEO, Dezful, Iran. [10] Present address: Department of Plant Biology and Ecology, University of the Basque Country (UPV/EHU), Apdo. 644, 48080 Bilbao, Spain. [11] Present address: Université de Bordeaux, 146 rue Leo-Saignat, Bordeaux, Cedex 33076, France. [12] Present address: Inari Agriculture, One Kendall Square Building 600/700, Cambridge, MA 02139, USA. [13] Present address: Centro de Biotecnología y Genómica de Plantas (CBGP, UPM-INIA), Universidad Politécnica de Madrid (UPM) - Instituto Nacional de Investigación y Tecnología Agraria y Alimentaria (INIA). Campus de Montegancedo-UPM, 28223-Pozuelo de Alarcón Madrid, Spain. [14] Present address: Wageningen University (Laboratory of Phytopathology), 6700AA Wageningen, The Netherlands. ✉email: cyrille.saintenac@inrae.fr

Crop immunity represents one of the most economically, sustainably, and ecologically relevant disease management strategies. Plant innate immunity was initially described as a two-layered system known as pattern (PAMP)-triggered immunity (PTI) and effector-triggered immunity (ETI)[1]. Since then, other models referred as effector-triggered defense (ETD), the invasion model and its simplified version (the spatial invasion model) were described in accordance to recent discoveries and especially in response observed against apoplastic leaf pathogens[2–4]. Genes involved in the control of these pathways belong to some of the largest gene families present in plant genomes. The cell-surface localized receptor-like kinases (RLK) and receptor-like proteins (RLP) are divided into several different sub-families including the leucine-rich repeat RLK (LRR-RLK) and the wall-associated kinases (WAK) among which major resistance genes/QTL have been identified[5]. The third largest class contains nucleotide-binding and leucine-rich repeats domains (NLRs). Breeding programs rely mostly on many of these genes that confer strong gene-for-gene type of resistance. However, they tend to break down quickly and have a short lifespan against evolutionary dynamic pathogens in monoculture cropping systems[6,7]. Species non-specific or race non-specific broad-spectrum resistance (BSR) genes and/or quantitative resistance represent alternatives for successful long-term resistance[8–12].

Wheat is grown on more land than any other crop worldwide, and wheat-based foods provide a significant source of calories to human kind. However, wheat production is threatened in many areas by devastating pests and pathogens[13]. The foliar disease Septoria tritici blotch (STB) caused by the hemibiotrophic and apoplastic fungal species Zymoseptoria tritici substantially impacts yield in some of the largest wheat-producing areas[13,14]. Resistance to fungicides[15] and the obligation to reduce fungicide treatments by 50% in some European countries (www.agriculture.gouv.fr, UE 2009/128/CE directive) poses a risk for maintaining high and durable wheat yields. Therefore, improvement of host resistance against STB in cultivated wheat is a high priority. To date, 22 major STB resistance genes (named Stb) have been mapped genetically in cultivated wheat, landraces, wild wheat species, and synthetic hexaploid wheat (SHW)[16,17]. Very recently, the cloning of the first Stb gene (Stb6) showed that it encodes a WAK that confers a gene-for-gene type of resistance against isolates carrying the matching AvrStb6 gene[18–20]. While this breakthrough paves the way for functional characterization of the molecular mechanisms involved, STB broad-spectrum and quantitative resistances yet need to be discovered to develop wheat cultivars with more durable resistance to this devastating disease.

In this work, we clone the Stb16q gene, which confers broad-spectrum resistance to Z. tritici. Using a TILLING population and stable transformation into a susceptible wheat accession we show that Stb16q encodes a cysteine-rich receptor-like kinase (CRK). We demonstrate that this gene, which localized at the plasma membrane stops pathogen growth very early in its infection cycle. Likewise, we show that Stb16q allele conferring resistance is nearly absent from the wheat hexaploid germplasm (excluding SHW) and was most likely recently brought by breeders into modern wheat varieties.

## Results

**Stb16q fine mapping reveals two candidate genes**. The bi-parental mapping population issued from the cross with Stb16q donor accession M3 presents a lack of recombination events in the Stb16q region that hampers any map-based cloning approach[21]. We showed that SHW TA4152-19 as well possess a unique broad-spectrum resistance against 64 geographically diverse Z. tritici isolates (Supplementary Data 1). Furthermore,

QTL analysis of TA4152-19 resistance using a bi-parental doubled haploid (DH) mapping population against two Z. tritici isolates showed the presence of a major resistance gene explaining up to 68% of the phenotypic variation located at the Stb16q locus (Supplementary Fig. 1). This confirmed the presence of Stb16q or a very closely linked gene in accession TA4152-19. To initiate the cloning of this gene, we mapped the flanking simple sequence repeat markers barc323 and cfd9 identified from the DH population on the high-density reference genetic map[22]. The marker interval showed the presence of 42 single nucleotide polymorphism (SNP) markers delineating 26 scaffolds of the D-genome physical map[23]. Five SSR markers designed from two of these scaffolds (scaffold1066 and scaffold45,305) co-segregated with Stb16q in the DH mapping population (Supplementary Fig. 2). New markers designed from both scaffolds were used to search for recombination events among 1305 immune and 675 highly susceptible F2 plants selected from a large-scale phenotyping experiment of 9100 F2 plants which allowed us to delimit Stb16q to a 0.07 cM interval between markers cfd335 and cfn80033 (Fig. 1). The screening and sequencing of TA4152-19 BAC clones and physical anchoring of Stb16q-flanking markers on the wheat genome reference sequence[24] revealed both the presence of only two candidate genes in this 272 kb interval, Crk6 and Unk1 (Fig. 1).

**Stb16q encodes a cysteine-rich receptor-like kinase**. To functionally validate these two candidate genes a set of complementary approaches were used. First, both candidate genes were sequenced from Stb16q-carrying accessions M3, TA4152-19 and the susceptible line ND495. Both SHWs shared the same haplotype for Crk6, which differed by 5% compared to the ND495 haplotype (Supplementary Fig. 3). Similarly, the same Unk1 haplotype was observed in both SHWs whereas the susceptible haplotype had a large insertion of 78 bp (Supplementary Fig. 3). Second, we evaluated the association of candidate gene haplotypes with STB resistance in a panel of 76 SHWs. Sequencing the Unk1 CDS (546 bp) and the genetically diverse Crk6 region corresponding to the first exon (853 bp) from the entire SHW panel revealed five and 11 haplotypes, respectively (Fig. 2 and Supplementary Fig. 3). All accessions carrying either Crk6 or Unk1 haplotypes from accession TA4152-19 were resistant to all Z. tritici isolates at the seedling stage (Supplementary Data 2). Third, we developed an EMS mutagenized population from TA4152-19 and phenotyped 310 M2 families with the Stb16q avirulent isolate IPO88018. This revealed the presence of two susceptible M2 plants from family 236. No mutation in the Unk1 CDS was observed for either of the M2 sister plants. However a point-mutation that resulted in an amino acid change from the conserved Ser to Phe at position 508 and predicted by PROVEAN[25] as deleterious was identified in the Crk6 kinase activation loop (Fig. 1, Supplementary Fig. 6, score = −5.617). Homozygosity of this mutation was associated with plant susceptibility in the segregating family 236 (Fig. 1 and Supplementary Data 3). Finally, the native genomic sequence of Crk6 (12.9 kb) was stably introduced into the susceptible French wheat cultivar Courtot. Regenerated T0 individuals positive for the presence of the transgene were self-pollinated and advanced to the T2 generation. Evaluation of T2 individuals with 11 Z. tritici isolates show that individuals carrying the transgene were resistant to all isolates (Fig. 1 and Supplementary Table 1). Altogether, these results demonstrated that the Crk6 gene is Stb16q, a cysteine-rich receptor-like kinase (CRK), which confers broad-spectrum resistance to Z. tritici.

**Stb16q stops Z. tritici at early infection stage**. The Stb16q gene contains seven exons and six introns with a transcript length of 3975 bp determined by PCR amplification of cDNA ends and

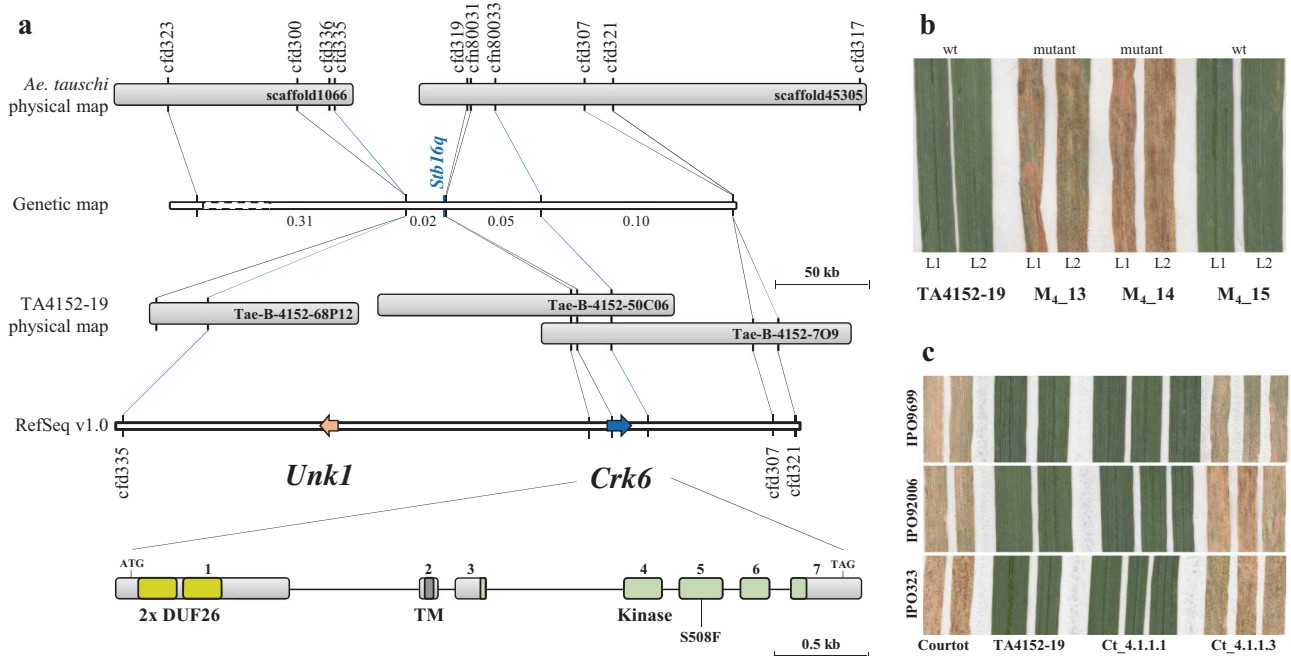

**Fig. 1 Map-based cloning of the Stb16q gene. a** Scaffold1066 and Scaffold45305 from the *Ae. tauschii* physical map[23] were identified as co-segregating with *Stb16q* in the DH population derived from TA4152-19 × ND495. Polymorphic SSR markers and SNP markers identified from gene/pseudogene fragments of these two scaffolds were genetically mapped based on 1980 $F_2$ individuals (distances in centiMorgans). This delimits *Stb16q* to a 0.07 cM interval between markers *cfd335* and *cfn80033*. Screening a TA4152-19 BAC library and the wheat reference sequence from Chinese Spring (RefSeq v1.0) identified a 272 kb interval with two candidate genes (scale bar = 50 kb). *Unk1* and *Crk6* are depicted as orange and blue arrows, respectively. A magnification of the *Crk6* structure reveals seven exons indicated as light gray boxes (scale bar = 0.5 kb). Yellow, dark gray, and green boxes represent the predicted DUF26 domains, the transmembrane domain, and the serine/threonine protein kinase domains, respectively. Mutation identified in the EMS-derived mutant family 236 is represented on the 5th exon. **b** Identification of the susceptible mutant family 236 derived from an EMS treatment of TA4152-19 seeds and carrying the deleterious mutation S508F. Phenotypes of leaf 1 (L1) and leaf 2 (L2) of three individual $M_4$ plants ($M_4\_13$, $M_4\_14$, and $M_4\_15$) from family 236 are represented 21 days after inoculation with *Z. tritici* isolate IPO88018. Mutation genotyping with marker *cfn80052* reveals that susceptible $M_4$ plants carry the mutation at the homozygous state while the resistant $M_4$ plants present the wild-type (wt) allele. **c** The native sequence of the *Crk6* gene including promoter and terminator (12.9 kb) was introduced via biolistic into the susceptible French wheat cultivar Courtot. $T_2$ plants from family Ct_4.1.1.1 carrying the *Crk6* transgene and its sister line Ct_4.1.1.3 lacking *Crk6* present resistant and susceptible phenotypes, respectively, on leaf 2 against three *Z. tritici* isolates (IPO9699, IPO92006, and IPO9699) at 21 dpi.

RNA-seq data. Similar to *Stb6*[18], expression of *Stb16q* is mainly observed in wheat leaves and increases along with the plant developmental stages (Supplementary Fig. 4). Furthermore, its expression is significantly upregulated in leaves of the resistant accession TA4152-19 as soon as one day after inoculation with the *Stb16q* avirulent *Z. tritici* isolate IPO09415 and in the susceptible accession ND495 with a peak prior the switch to the necrotrophic phase (Supplementary Fig. 5). *Stb16q* encodes a cysteine-rich receptor-like kinase of 684 amino acids with two extracellular copies of the DUF26 domain containing conserved C-X8-C-X2-C motifs, a predicted transmembrane domain, and an arginine-aspartate (RD) intracellular serine/threonine (Ser/Thr) protein kinase domain (Fig. 1 and Supplementary Fig. 6). Transient expression assay by agroinfiltration of *Nicotiana benthamiana* leaves showed colocalisation of the STB16-GFP fusion protein with AtFH6 protein, a plasma membrane-associated protein used as a marker[26] confirmed its predicted localization at the plasma membrane (Supplementary Fig. 7). All amino acids involved in mannose-binding activity of the DUF26-containing protein *Gnk2*[27] are conserved in STB16 C-terminal DUF26 domain (Supplementary Fig. 6) suggesting that STB16 may recognize apoplastic plant or fungi-derived mannose or derivatives to trigger *Stb16q*-mediated broad spectrum defense. Once initiated, this mechanism stops the progression of the pathogen either before its penetration through the stomata or into the sub-stomatal cavities as illustrated by

confocal microscopy study of *Stb16q* near-isogenic lines infected with GFP-labeled *Z. tritici* isolate 3D7 (Fig. 3). As shown in other incompatible interactions[3], this mechanism does not kill the fungus (Supplementary Fig. 5).

**Stb16q was recently introduced into wheat varieties**. *Stb16q* exon1 resequencing from 156 accessions (including 73 SHWs and cultivars from 29 countries) revealed 12 highly diverse haplotypes (Fig. 2, Supplementary Data 4). The resistant haplotype was observed only in SHWs and most of the *Stb16q* genetic diversity (ten out of the 12 haplotypes) was identified among the SHWs. A unique susceptible haplotype was observed in all sequenced hexaploid wheat lines (excluding SHWs) except in the landrace Oznaka Linije I. Screening of 372 accessions from a core-collection known to encompass maximum wheat genetic diversity[28] and 439 cultivars/advanced lines from the main wheat producing areas (total of 805 accessions originated from 61 countries) with the *Stb16q* diagnostic SNP markers *cfn80044* and *cfn80045* indicated that the resistance allele is absent from the core collection and present in only five wheat cultivars and one advanced breeding line (Supplementary Data 5). Further screening of 136 *Ae. tauschii* accessions shows that *Stb16q* likely originated near the Caspian Sea and belong exclusively to *Ae. tauschii* genetic lineage 2 (Fig. 2, Supplementary Data 6)[29]. Altogether, these results suggest that *Stb16q* was brought recently

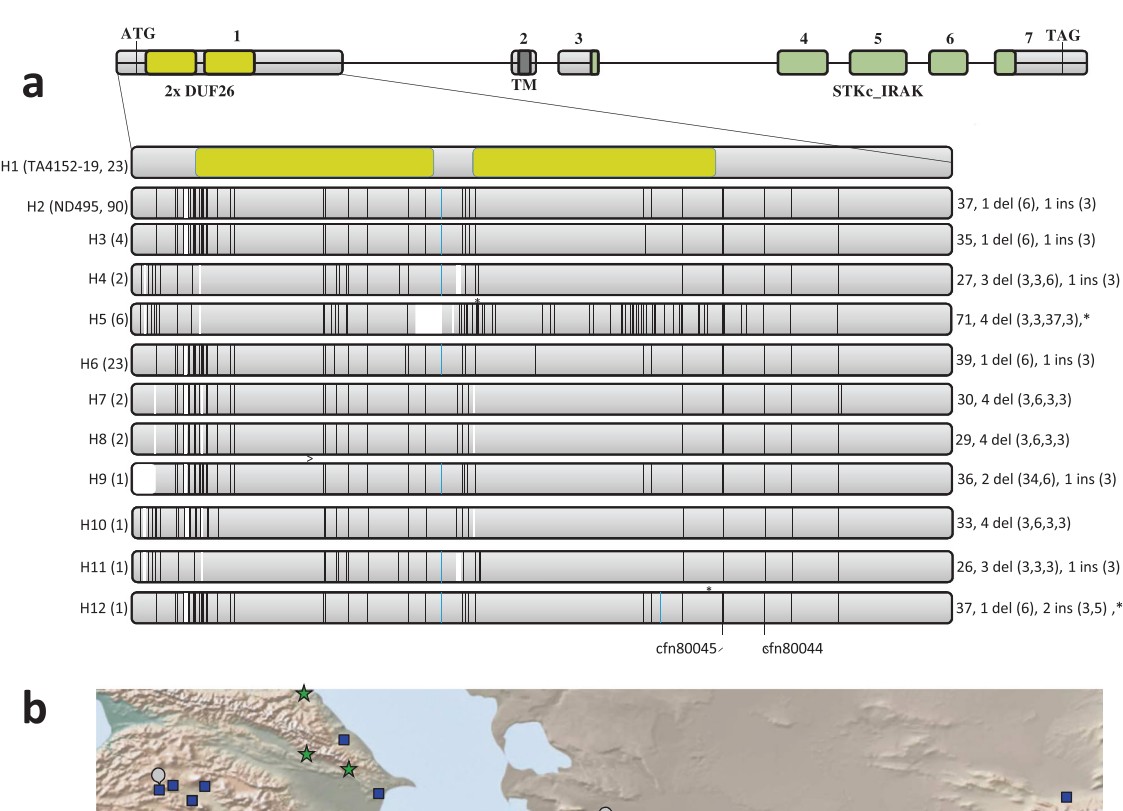

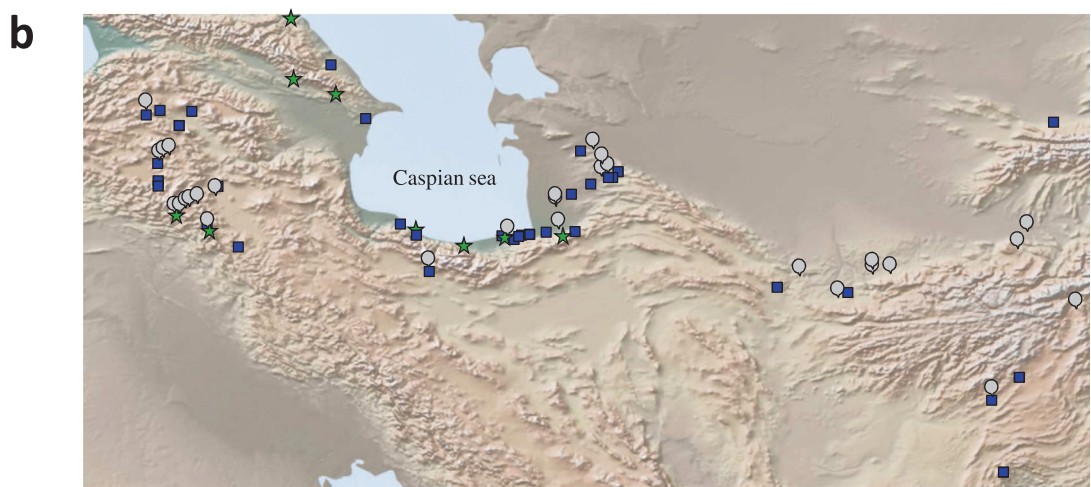

**Fig. 2 Stb16q genetic diversity and origin. a** Schematic representation of the 12 *Stb16q* exon1 haplotypes (H1 from H12) identified from re-sequencing 156 wheat accessions (including the 76SHWs, Supplementary Data 4). *Stb16q* structure reveals seven exons indicated as light gray boxes. Yellow, dark gray, and green boxes represent the predicted DUF26 domains, the transmembrane domain, and the serine/threonine protein kinase, interleukin-1 receptor associated kinase (STKc_IRAK) domains, respectively. Vertical black, white, and blue bars represent SNP, deletions, and insertions, respectively, compared to haplotype 1. On the left of each haplotype are indicated in brackets the most relevant representative and the number of accessions identified for each haplotype. On the right of haplotypes are indicated the number of SNP, number of deletions (their size in bp), and number of insertions (their size in bp). Asterisks highlight the presence of a stop codon. Physical position of diagnostic markers *cfn80044* and *cfn80045* are indicated at the bottom. All haplotypes have been identified in the SHWs collection except haplotype H12. **b** Topographic distribution of *Ae. tauschii* accessions carrying the *Stb16q* gene (green star) or not (blue square) and undetermined allele (gray round) according to diagnostic markers *cfn80044* and *cfn80045* (Supplementary Data 6). The map was created using the GPS Visualizer software (https://www.gpsvisualizer.com/about.html).

into wheat varieties through the use of synthetic wheat in breeding programs further highlighting the importance of genetic resources for wheat improvement[30].

**Stb16q does not confer susceptibility to other diseases**. Disease resistance genes are valuable for breeding if they do not lead to trade-offs to other pathogens[31]. We evaluated the impact of *Stb16q* against other wheat fungal diseases. Screening of Chinese Spring NILs (BC$_5$F$_2$) plants segregating for *Stb16q* for reaction to Septoria nodorum blotch (SNB, three isolates), tan spot (two isolates), and stem rust (two races) and genotyping them with

marker *fcp722* indicated that *Stb16q* does not confer susceptibility to these other major wheat pathogens (Supplementary Table 2).

## Discussion

Plant resistance to pathogens depends on a number of complex large gene families such as nucleotide-binding leucine-rich repeat (NLR) and receptor like kinase (RLK). Here, we show that *Stb16q* encodes a member of the cysteine-rich receptor-like kinase RLK sub-family. The number of cloned resistance genes has steeply increased over the last five years[5]. While the nature of these genes is diverse, none of them so far belongs to the CRK family.

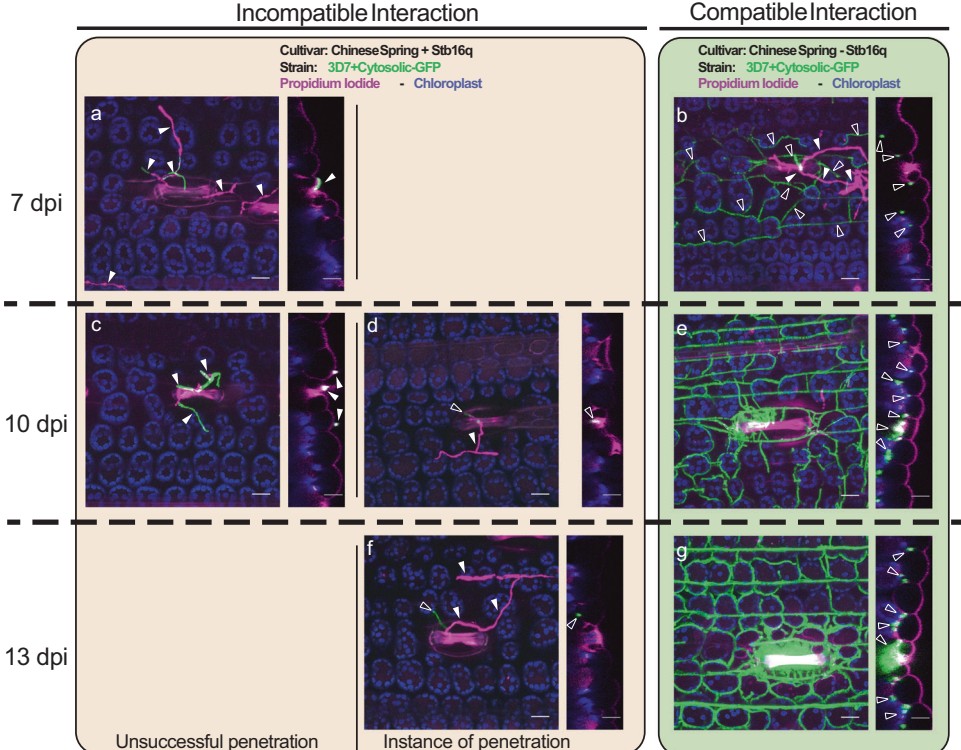

**Fig. 3 Zymoseptoria tritici infection during compatible and incompatible interactions using wheat near isogenic lines carrying the Stb16q gene.** Leaf surfaces and leaf cross sections of Chinese Spring carrying or not *Stb16q* were observed using a confocal at three time points after infection (7, 10, and 13 dpi) with the *Stb16q* avirulent GFP-labeled *Z. tritici* isolate 3D7. Epiphyllous and penetrating hyphae, shown in green or in purple when they were stained by propidium iodide, are marked by full and empty white triangles, respectively. At 7 dpi, in the incompatible reaction (CS with *Stb16q*), the fungus is observed only at the leaf surface and no instance of penetration was recorded (**a**). At 10 dpi, some instances of penetration were observed (**d**) and by 13 dpi, hyphae start growing inside the substomatal cavity (**f**). In the compatible interaction (CS without *Stb16q*), by 7 dpi the fungus hyphae colonized the mesophyll (**b**). By 10 and 13 dpi, it formed a dense network and began to fill the substomatal cavities and to form pycnidia (**e**, **g**). Images **a**–**g** are maximum projections. The corresponding orthogonal views (*yz*) are shown in the narrow panels to the right of each main image. Scale bars represent 20 μm. The autofluorescence of the chloroplasts is shown in purple and the cell walls stained with propidium iodide in pink. The experiment was performed twice (*n* = 2). In the first replicate, infected leaves were collected at 6 and 9 dpi while they were collected at 7, 10, and 13 dpi in the second replicate. Both replicates show the same result.

Recently, the cotton CRK *GbCRK18* was shown to play a role in resistance to *Verticillium dahliae* as silencing this gene by virus-induced gene silencing mediates plant susceptibility[32]. However, it is not known if this gene would confer resistance once introduced into a susceptible accession. Members of this large multi-gene family have been identified in other crops as candidate genes for resistance against different pathogens[33–35]. These data emphasize the potential involvement of another class of major genes in crop diseases resistance. Members of the CRK sub-family contain an extracellular domain comprising generally two DUF26 domains (also named Gnk2 or stress-antifungal domain and harboring a conserved C-X8-C-X2-C motif) involved in ligand recognition, a single-pass transmembrane domain, and a conserved intracellular serine/threonine (Ser/Thr) protein kinase domain needed for intracellular signal transduction. The 44 members of this family have especially been well studied in *Arabidopsis* and they play a role in many different biological processes including resistance against pathogens[36]. Their involvement in many different physiological responses and the absence of *Stb16q* orthologue prevent any extrapolation of *Stb16q* mode of action.

*Stb16q* is the second *Stb* genes cloned after *Stb6*, which encodes a wall-associated-like receptor kinase[18]. The cloning of the first two *Stb* genes highlights the importance of wheat RLKs in resistance to STB and it is not speculative to say that some of the other *Stb* genes should belong to this large gene family. RLK are known to perceive molecular compounds present in the apoplasm such as pathogen-associated molecular patterns (PAMPs), damage-associated molecular patterns (DAMPs), or avirulence proteins to trigger a signaling cascade through the activation of their kinase domain. Both *Stb* genes belong to different RLK sub-families, which suggest that the molecular mechanism involved in resistance is different. At least, they should recognize different apoplastic molecular compounds as they carry different extra-cellular domains. *Stb6* ligand is still unidentified but it is known that WAKs bind oligogalacturonides[37]. The *Stb16q* extracellular DUF26 domains are present in three different classes of proteins, cysteine-rich receptor-like secreted proteins (CRRSPs), cysteine-rich receptor-like kinase (CRKs), and plasmodesmata-localized proteins (PDLPs). GNK2 from *Ginkgo biloba* and maize AFP1 CRRSPs bind mannose[27,38]. No ligand was identified for PDLPs and CRKs[39]. However, one of the *Stb16q* DUF26 domains shares the same three amino acids required for GNK2 mannose binding, which suggests a potential affinity of *Stb16q* DUF26 domain for this molecule. Altogether, this suggests that the apoplastic molecular composition, around the living space of *Z. tritici*, and especially carbohydrates diversity appear critical for determining the fate of the interaction between the fungus and the plant.

Broad-spectrum resistance when dealing with genetically diverse pathogens is particularly of interest for breeders. No fully virulent *Z. tritici* isolate was identified against SHWs TA4152-19 and M3, which makes them as priority targets for improving wheat resistance to STB. According to genetic analyses, *Stb16q* explained most of the phenotypic variability in bi-parental population issued from these accessions[21]. Therefore, *Stb16q* most likely explains their broad-spectrum resistance even if we cannot rule out that these accessions carry other resistances against *Z. tritici*. There are several reasons that may explain *Stb16q* broad-spectrum resistance either its molecular nature and/or its resistance mechanism or its history. We showed that this gene encodes a CRK, which may recognize a carbohydrate to trigger the resistance mechanism. Recognition of a carbohydrate conserved among the *Z. tritici* population or systematically produced during wheat infection by *Z. tritici* could explain this broad-spectrum resistance. We also showed that *Stb16q* was almost absent in the cultivated wheat varieties. This means that it did not apply any selection on the pathogen population. This could explain the absence of virulence among the tested isolates, which were all sampled more than five years ago. While it is difficult to conclude as very little is known on CRK proteins, *Stb16q* BSR is most likely due to its nearly absence from the field. There are very few examples of genes that confers BSR while present in cultivated varieties and most of those involved in BSR are new genes originated from wild species[6]. This means that this gene with the help of the diagnostic markers should be carefully deployed with other qualitative or quantitative resistance to breed for durable broad-spectrum resistant wheat varieties against the devastating STB disease.

## Methods

**Plant materials and phenotypic evaluation.** Segregating populations consisting of 170 doubled haploid (DH) lines and $F_2$s derived from a cross between the synthetic hexaploid wheat line TA4152-19 (*Stb16q*) and the susceptible hard red spring wheat line ND495, were used for genetic linkage mapping. Chinese Spring (CS) near-isogenic lines (NILs) segregating for *Stb16q* were obtained following five backcrosses starting with $F_1$ CS × TA4152-19. At each generation, plants were phenotyped with *Z. tritici* isolate IPO9415 and genotyped with SSR markers *cfd306* and *cfd323* to maintain *Stb16q* in the progenies. A $BC_5F_1$ plant heterozygous at the *Stb16q* locus was self-pollinated and segregating $BC_5F_2$ plants were used for phenotyping against different pathogens. Genotyping of two sister $BC_5F_2$ NILs (*Stb16q* and *stb16q*) identified with marker *cfn80044* with the Axiom® 35k breeders SNP array[40] confirmed the status of *Stb16q* and revealed a 6.9% of residual heterozygosity per line. All genotyping CEL files from the GeneTitan were analyzed with the APT suite. The wheat core-collection of 372 accessions was provided by the Biological Resource Center on small grains cereals (INRA, France)[28]. The *Ae. tauschii* collection and the synthetic wheat panel were obtained from the National Small Grains Collection (USDA-ARS, Aberdeen, Idaho, USA) and the Wheat Genetics Resource Center (KSU, Manhattan, Kansas, USA), respectively. All these accessions and the wheat cultivars collection originated worldwide are further described in Supplementary Data 5 and 6.

Segregating populations and EMS-derived families were evaluated at PRI under the protocol previously described[21]. Evaluation of the SHW collection was performed at INRA. Three plants per one liter pots filled with ½ blond and ½ brown peat mosses were paintbrush inoculated with a $10^6$ solution of spores per ml on the second leaves 14 days after sowing. The plants were kept under 100% humidity for 48 h and then grown under 85% humidity, 16 h daylight period, and 21/18 °C (day/night). Symptoms were recorded visually around 21 dpi.

**Genetic mapping.** A complete list of primers used is supplied as Supplementary Data 7. In the first phase of mapping and analysis, 30 polymorphic simple sequence repeat (SSR) markers and six previously developed sequence-tagged site (STS) markers[41] (Supplementary Fig. 1) were used to genotype the population of 170 DH lines, and a genetic linkage map of chromosome 3D was assembled as previously described[21]. Over 300 bin-mapped expressed sequence tagged (EST) sequences that were known to map to the most distal deletion bin of 3DL (containing the *Stb16q* locus) were then downloaded and used to develop PCR primers, which were tested on TA4152-19 and ND495 for polymorphism and used to generate a genetic linkage map. Regression mapping was used to identify markers significantly associated with STB resistance (Supplementary Fig. 1). In a second phase, we used the ITMI (175 DH, 4,209 markers + 416,856 GBS markers) reference genetic maps[22,42] to increase further the number of markers at the *Stb16q* locus.

Polymorphic SSR markers *barc323* and *cfd9* flanking *Stb16q* on the DH TA4152-19 × ND495 population were used to genotype the ITMI population and analyzed using the GenMapper software. Using the RQTL package and Carthagene[43], we constructed draft 3D linkage maps that included the *Stb16q* flanking markers. In parallel, based on the reference map containing 4209 markers, we extracted individuals with recombination events located closely to the *Stb16q* region. Based on these individuals, we selected all markers with the same genotype as markers *barc323* and *cfd9* as well as markers that might be in between according to the distribution of recombination events. A total of 42 genotyping-by-sequencing (GBS)-derived SNP markers originating from 33 short GBS tags were identified. These 33 GBS tags were subjected to BLAST searches against D genome sequences[23,44]. A total of 11/33 GBS sequences matched to 10 scaffolds anchored to the linkage map produced by Jia et al.[23]. Comparisons between the 3D linkage map and the physical map indicated that the *Stb16q* region (between *barc323* and *cfd9*) corresponds to a physical region containing 26 scaffolds (from scaffold51510 to scaffold39406, Supplementary Fig. 2). No match was identified with the second dataset[44] that contained mostly short genic sequences. In order to use the high number of SNP markers developed in this second dataset, we aligned the short sequences to the 26 large scaffolds surrounding the *Stb16q* locus. We identified six short sequences that matched one of the large scaffolds. We developed 18 SNPs KASP assays for each of the SNPs from the 10 K assay present in the region and tested them against the parents of the mapping populations. Only three of them (*cfn80007*, *cfn80011*, and *cfn80016*) were genetically mapped using the bi-parental DH mapping population (Supplementary Fig. 2).

SNPs genotyping was achieved with the KASP™ genotyping chemistry according to manufacturer instruction (LGC group) in an 8.11 μl final volume and analyzed on the LightCycler® 480 Real-Time PCR System (Roche Life Science). In addition, using SSR Locator[45], we searched for all SSRs present on the 26 large scaffolds. Using the Primer3Plus software, primers were designed for 15 SSRs containing dinucleotide and mononucleotide motifs with a high number of repetitions and tested on individuals TA4152-19, ND495, ditelosomic 3DS/L, deletion line 3DL-3[46], and the DH population. SSR genotyping was performed as described previously[47]. A total of 10 SSR markers were genetically mapped using the DH population (Supplementary Fig. 2). This allowed restricting *Stb16q* locus to two scaffolds (scaffold1066 and scaffold45305). These latter were analyzed further to design primers for all SSRs identified. Genotyping CS aneuploid lines reveals that three SSR markers were assigned to the 3D short arm. Scaffold45305 was then mostly likely miss-assembled and only the region assigned to the 3D chromosome was used for further analysis.

In addition, re-sequencing gene fragments from scaffold45305 revealed two SNPs between TA4152-19 and ND495 used to develop SNP markers *cfn80031* and *cfn80033*. Genotyping the DH population revealed five SSR markers (*cfd300*, *cfd307*, *cfd317*, *cfd319*, and *cfd321*) that co-segregated with *Stb16q* on the DH population. To carry on the fine-mapping, 1305 fully resistant ($P = 0$) and 675 fully susceptible ($P > 40$) $F_2$ individuals were genotyped with *Stb16q*-flanking markers *cfd306* and *cfd323*. A total of 87 individuals carrying a recombination event were genotyped with all available markers present in the *Stb16q* region. The 20 individuals carrying a recombination event between markers *cfd300* and *cfd307* were self-pollinated and progenies were phenotyped with isolate IPO88018 to determine *Stb16q* genetic status. Genetic distance for a given interval is represented by the percentage of recombined gametes.

**Stb16q physical map.** A wheat BAC library designated Tae-B-4152-ng was constructed from genomic DNA of wheat genotype TA4152-19, at the Centre National de Resources Génomiques Végétales (CNRGV)- INRAE, France. The non-gridded BAC library was constructed based on the protocol described previously[48] with the modifications described in Mago et al.[49]. High molecular weight (HMW) DNA was prepared from 20 g of leaves. Embedded HMW DNA was partially digested with HindIII (Sigma-Aldrich, St-Louis, Missouri), size selected, eluted, and ligated into pIndigoBAC-5 HindIII-Cloning Ready vector (Epicentre Biotechnologies, Madison, Wisconsin). Two successive HMW DNA extractions and three independent sizing steps led to production of 456 000 BAC clones. To evaluate the average insert size of the library, the DNA was isolated from randomly selected BAC clones, digested with the NotI restriction enzyme, and analyzed using pulsed-field gel electrophoresis. All fragments generated by NotI digestion contained the 7.5-kb pIndigoBAC-5 vector band and various insert fragments. Analysis of insert sizes from these BAC clones indicated a mean average size of 108 kb. Thus, the BAC clones produced represent a total of ~3.3-fold coverage of the genome. BAC clones were divided into 320 pools before overnight growth and DNA amplification.

Pools were screened using *Stb16q* linked PCR markers *cfd300*, *cfd336*, *cfd319*, *cfn80031*, and *cfd307*. Three positive BAC clones (7O9, 50C06, and 68P12) were identified and sequenced using the PacBio RS II sequencing system (Pacific Biosciences). Two μg of BAC clone DNA were pooled with 11 other BAC clones DNA to obtain a total amount of 24 μg. One library was generated using the standard Pacific Biosciences library preparation protocol for 8–12 kb libraries. This library was sequenced in one PacBio RS II SMRT Cell using the P6 polymerase in combination with the C4 chemistry (sequencing service following the standard operating procedures was provided by IGM Genomic Center). Assembly of the PacBio RS II reads was performed following the HGAP workflow. The SMRT® Analysis (v2.2.0) software suite was used for HGAP implementation

(https://github.com/PacificBiosciences/Bioinformatics-Training/wiki/HGAP). Reads were first aligned using BLASR[50,51] against "Escherichia coli str. K12 substr. DH10B, complete genome". Identified E. coli reads and low-quality reads (read quality <0.80 and read length <500 bp) were removed from data used for the BAC clone sequences assembly. Vector sequences were trimmed along the assembly process. Each BAC assembly was individualized by matching their BES on the ends of assembled sequences with BLAST.

Sequences were annotated automatically using the TriAnnot pipeline[52] and Stb16q annotation was curated manually according to expression data.

To fill in the physical gap, depth of coverage of the Tae-B-4152-ng BAC library was increased twice. Each time, a one-fold coverage of the genome was achieved. An ISBP marker (cfp50029) designed at the 5′ end of BAC clone 50C06 (Fig. 1) was used to screen the BAC library. No other BAC clone was identified. In addition, sequences of PCR primers and gene models identified from BAC clones were searched on the Chinese Spring whole genome assembly (RefSeq v1)[24] by BLAST search. A region of 353 kb between markers cfd335 and cfd321 was identified on chromosome 3D. The two gene models (TraesCS3D01G500700, TraesCS3D01G500800) of this interval were retrieved from the IWGSC annotation and compared with the manually curated annotation performed on the TA4152-19 sequences.

**Mutagenesis of the SHWs TA4152-19.** Seed of the synthetic hexaploid wheat line TA4152-19 was obtained from the Wheat Genetics Resource Center, Kansas State University, Manhattan, KS, USA. A total of 607 TA4152-19 seeds were treated with 0.35% ethyl methane-sulfonate (EMS) in 0.05 M phosphate buffer as previously described[53]. After treatment, the seeds were directly sown in 6.9 × 25.4 cm cones (Stuewe and Sons, Inc. Tangent, OR) containing SB100 Professional Growing Mix (Sungrow Horticulture, Bellevue, WA) in a greenhouse with an average temperature of 20–23 °C and a 16-h photoperiod. The M$_1$ plants were allowed to self-pollinate and the M$_2$ seed harvested. A total of 295 plants either did not germinate, died, or were sterile and did not produce seed.

About 20 M$_2$ plants of the remaining 310 families were inoculated with isolate IPO88018 according to the procedure described above. Two plants from family 236 were identified as susceptible. Crk6 and Unk1 CDS were sequenced from both plants using Sanger sequencing. A point mutation at position 1523 (C to T) was identified in Crk6. The functional impact of this mutation was analyzed using PROVEAN[25] (score = −5.617) and the consensus classifier PredictSNP[54] (87% confidence). Heterozygous M$_2$ and M$_3$ plants at this position were allowed to self-pollination. A total of 54 M$_4$ plants (M4_TA4152-19_236_C1) derived from heterozygous M$_3$ plants were phenotyped with the Stb16q avirulent isolate IPO88018. Mutation genotyping was performed using the KASP$^{TM}$ genotyping chemistry (LGC Genomics) and primers cfn80052 (Supplementary Data 3) and analyzed on the Light Cycler 480 Real-Time PCR system (Roche Applied Science).

**Complementation.** A genomic sequence of 12.9 kb containing the full length Crk6 gene was PCR amplified from the BAC clone 7L08 using the Phusion High-Fidelity PCR Master Mix (Thermo Fisher Scientific) and primers Crk6F11 / Crk6R7. Single deoxyadenosine was added to the 3′ ends of the PCR product using the Taq DNA polymerase from New England Biolabs. The PCR product was cloned into the pCR8/GW/TOPO vector (Thermo Fisher Scientific). Integrity of the cloned genomic sequence was verified by Sanger sequencing of PCR fragments produced with a set of primers pairs distributed along the Stb16q gene sequence. The pCR8/GW/TOPO vector carrying Stb16q was linearized with BssHII and the cloned wheat genomic DNA fragment purified following an ethanol precipitation and dephosphorylated as previously described[55]. This fragment was then mixed together with the 'bar dephosphorylated cassette' at a 2:1 ratio and used for transformation of immature embryos of wheat Courtot by particle bombardment using a PDS 1000 He device (BioRad). Regeneration of plants and bar selection were performed essentially as previously reported[55].

Detection of the Stb16q gene in T$_0$ plants was performed by PCR amplification using plant genomic DNA as template and primers 20F6 and 20R7. Two independent positive transgenic lines (Ct_4.1.1.1 and Ct_4.2.1.3) were identified. T$_0$ plants were allowed to self-pollinate and the resulting T$_1$ progeny was assessed for resistance to Z. tritici IPO323. Resistant individual plants were self-pollinated and T$_2$ progenies were assessed for resistance to different Z. tritici isolates at seedling stage. The sister T$_2$ family Ct_4.1.1.3 originated from the same T$_0$ plant as family Ct_4.1.1.1 but from a different tiller was used as a negative control.

**Expression of Stb16q.** Leaf fragments of accessions TA4152-19 and ND495 were collected at 0, 1, 2, 5, 10, 14, 17, and 21 days after paintbrush inoculations with either water or Z. tritici isolate IPO9415. Three independent biological replicates (three leaves each) were analyzed. Total RNA was isolated from leaf tissues using the TRIzol® reagent (Invitrogen) and then treated with DNAse (TURBO DNA-free™ Kit, Life technologies). First-strand cDNAs were synthesized from 1 µg of total DNA-free RNA primed with oligo d(t) using the iScript™ Select cDNA Synthesis kit (Bio-Rad). qRT-PCR experiments were performed using the Light-Cycler®480 SYBR Green I Master (Roche) and the LightCycler® 480 Instrument (Roche) under the following thermal profile: initial denaturation at 95 °C for 10 min; followed by 40 cycles of 15 s at 95 °C, 15 s at 62 °C, and 15 s at 72 °C;

and final extension at 72 °C for 15 s. Primers efficiencies were obtained using five four-fold dilutions of cDNA. Relative expression of target genes Crk6 (20F1/CRK6_R1) and β-tubulin (MgTub_F/R) were calculated according to the Pfaffl method[56] (control = samples at T$_0$) and normalized by the Phytochelatin[57] (PhytoF/R1).

Rapid Amplification of cDNA Ends (RACE) was performed using the SMARTer RACE 5′/3′ kit (Clontech) following the manufacturer's instructions to determine the 5′ and 3′ end of Crk6. TA4152-19 cDNA originated from all time points were pooled in equimolar concentration and used as template for 5′ and 3′ ends amplification using primers Crk6R1 and Crk6F1/20F1 (nested PCR), respectively. PCR products were cloned into the pRACE vector (Clontech) and inserts were sequenced using primers M13F and Crk6F8 at GATC Biotech SARL (Konstanz, Germany) for 5′ and 3′ ends, respectively. Sequences were compared to ends identified by the TriAnnot pipeline and the consensus was chosen as the initiation and termination sites of transcription. Expression of Crk6 at different developmental stages was evaluated using the Wheat eFP Browser[58] and the WheatExp pipeline[59].

**Subcellular localization of STB16 in N. benthamiana.** Stb16q CDS was PCR-amplified from first-strand cDNA derived from leaves of accession TA4152-19 collected 17 days after inoculation with Z. tritici isolate IPO9415. The reaction was performed using primers Stb16_CDS_AttB2_ACC and Stb16_CDS_AttB2_C-fus and the Phusion® High-Fidelity PCR Master Mix (Thermo Scientific™) with the following conditions: 95 °C for 10 min followed by 35 cycles of at 95 °C for 15 s, 69 °C for 30 s, and 72 °C for 2 min. The purified PCR product was cloned into the pDONR221 and then transferred to the pK7FWG2 expression vector using the Gateway Technology (Invitrogen). The AtFH6 exon 1 coding sequence[26], the P35S promoter, and the mCherry sequence were recombined into the pH7m34 expression vectors with Gateway Technology (Invitrogen). Both vectors were transferred into Agrobacterium tumefaciens and infiltrated in N. benthamiana leaves as previously described[60]. Leaf epidermal cells were imaged three days after agro-infiltration, with an inverted confocal microscope (model LSM 880; plan-apochromat 40x/1.4 Oil DIC M27 objective, Zeiss) equipped with an Argon ion and HeNe laser as the excitation source using the Zeiss ZEN2.3 SP1 (black edition) version 14.0.7.201. For simultaneous GFP/mCherry imaging, samples were excited at 488 nm for GFP and 543 nm for mCherry, in the multi-track scanning mode. GFP emission was detected selectively with a 505–530 nm band-pass emission filter. We detected mCherry fluorescence in a separate detection channel, with a 560–615 nm band-pass emission filter. Confocal images were finally processed using the ZEN 2.3 (blue edition) version 2.2.69.1000.

**Western blot.** Total proteins were extracted from agro-infiltrated N. benthamiana leaves using a protein extraction buffer (10% glycerol, 25 mM Tris pH7.5, 1 mM EDTA, 150 mM NaCl, 2% of polyvinylpolypyrrolidone, 0.5 mM DTT, 0.1% Tergitol) with a Plant Protease Inhibitor Mix (Sigma-Aldrich, St Louis, MO, USA). After centrifugation (10,000 g, 5 min), the supernatant is boiled in Laemmli buffer 2X (Bio-Rad Laboratories, Munich, Germany) supplemented with DTT and separated on 12% Tris-Glycine SDS-PAGE gel, transferred to PVDF membrane (Bio-Rad Laboratories). Immunoblots were blocked and incubated in 5% milk/PBS-T (PBS with 0.1% Tween 20) overnight at 4 °C with GFP antibodies at 1:5000 (#3H09 ChromoTek, Martinsried, Germany). After washes with PBS-T, blots were incubated in 5% milk/PBS-T, 45 min at room temperature with secondary antibodies at 1:10 000 (goat anti-rat IgG-HRP AS10 1187 Agrisera Sweeden). Proteins were visualized using the chemiluminescence detection kits, Luminata™ Forte Western HRP substrate (Merck Millipore, Watford, UK) following the manufacturer's instructions..

**Imaging Z. tritici infection in NILs carrying or not Stb16q.** Hyphae growth on the leaf surface was observed using an inverted Zeiss LSM 780 confocal microscope using Argon (488 nm) lasers as illumination sources. Images were acquired using the Zeiss Zen Black 2012. Signal detection for emissions were set as follows: eGFP (494.95–535.07 nm); chloroplast autofluorescence (656.01–681.98 nm); and propidium iodide (596.87–632.38 nm). The strain ST99CH3D7 (3D7) used for confocal microscopy was transformed by S. Kilaru and G. Steinberg with a codon optimized version of cytoplasmic eGFP[61]. 13–15-day old plants of Stb16q NILs were infected with the strain 3D7 expressing cytoplasmic eGFP at a concentration of 10$^6$ spores/ml, as described before[62]. The experiment was performed twice. In the first replicate, infected leaves were collected at 6 and 9 dpi. We observed that at 6 dpi there were no penetration in Stb16q-containing lines while at 9 dpi some events of successful penetration by Z. tritici through the stomata were observed. Leaves of the second experiment were collected at 7, 10, and 13 dpi. Three cm from the tip of the leaf were discarded and the adjacent 1.5 cm segments were incubated in the dark for 10 min in 15 µM (in H$_2$O 0.02% tween20) propidium iodide (Invitrogen). After rinsing, the leaf segments were mounted on slides with milliQ water + 0.02% Tween20. Fiji package[63] of ImageJ was used to adjust brightness and contrast, cropping and scale bar additions.

**Stb16q haplotypes analysis.** A total of 156 hexaploid wheat accessions were used to identify Stb16q haplotypes. This collection was composed of 73 SHWs, 45

accessions originated from the wheat core-collection[28], 23 recent cultivars, 12 accessions from the wheat *Stb* differential[16], ND495 and two landraces (Supplementary Data 4). *Stb16q* CDS was entirely sequenced only for accessions carrying the same exon1 as TA4152-19. PCR products were generated using the AmpliTaq Gold® Master Mix (Applied Biosystems), purified using Agencourt AMPure XP (Beckman Coulter), and Sanger sequenced at GATC Biotech SARL (Konstanz, Germany). The MEGA software was used to align the different sequences and to identify haplotypes.

**Development of STARP and KASP diagnostic markers**. The KASP markers cfn80044 and cfn80045 developed from SNPs at position 755 (counting from the translation start site in TA4152-19) (G/A) and 704 (G/A), respectively were used to screen the wheat worldwide collection and the *Ae. tauschii* collection. Reaction was performed following manufacturer instructions (LGC Genomics) and run on the Light Cycler 480 Real-Time PCR system (Roche Applied Science).

The two adjacent SNPs, including G/A and G/C at nucleotide positions 704 and 705 were targeted for STARP marker development following the methods outlined in Long et al.[64] and used to amplify genomic DNA of the BC$_5$F$_2$ plants used for evaluation of tan spot, stem rust, and SNB resistance following the protocols of Long et al.[64]. PCR products were electrophoresed on 6% polyacrylamide gels, stained with GelRed, and visualized on a Typhoon 9500 variable mode imager (GE Healthcare, Waukesha, WI, USA). This STARP marker fcp722 (*Stb16q*) revealed clear, codominant polymorphic fragments between the *Stb16q*-containing lines (TA4152-19 and M3) and lines lacking *Stb16q*.

**Evaluation of CS BC$_5$F$_2$ lines for reaction to other diseases**. Chinese Spring BC$_5$F$_2$ lines segregating for *Stb16q* were evaluated for reaction to multiple isolates/races of the pathogens that cause stem rust (*Puccinia graminis*), Septoria nodorum blotch (SNB) (*Parastagonospora nodorum*), and tan spot (*Pyrenophora tritici-repentis*) to determine if *Stb16q* confers resistance to other diseases. For stem rust evaluations, *P. graminis* races TPMK and QFCSC were inoculated on 23 and 26 BC$_5$F$_2$ seedlings, respectively, using the methods and conditions described in Faris et al.[65] and scored using the scale described by Stakman et al.[66]. SNB evaluations were carried out by inoculating *Pa. nodorum* isolates Sn4, Sn6, and Sn2000 onto 15, 17, and 13 BC$_5$F$_2$ plants, respectively. Plant growth conditions, inoculum production, inoculation procedures, and reaction type scores were as previously described[67]. Tan spot evaluations were conducted using *Py. tritici-repentis* isolates Pti2 (race 1) and DW5 (race 5). Each isolate was inoculated on to 25 BC$_5$F$_2$ plants following the methods and procedures described in Liu et al.[68] and scored using the lesion type scale developed by Lamari and Bernier[69].

**Statistical analysis**. To evaluate significant differences between *Stb16q* expression data and *Z. tritici* biomass during a time course infection on wheat carrying the resistant (TA4152-19) or the susceptible (ND495) *Stb16q* haplotype, we applied the non-parametric Van der Waerden test using the RStudio environment. Group containing the same letter do not differ significantly at $P < 0.05$ according to the khi2 test.

**Reporting summary**. Further information on research design is available in the Nature Research Reporting Summary linked to this article.

## Data availability
The source data for Supplementary Figs. 5 and 7, Supplementary Data 1, 2 and Supplementary Tables 1 and 2 are provided as a source data file. *Stb16q* haplotypes and BAC clones sequences are available as GenBank accession numbers MT231554, to 231565, MT932490, MT942596, and MT942597. BAC clones raw sequencing data are available as SRA number PRJNA680685. Source data are provided with this paper.

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

## Acknowledgements

We dedicate this work to the memory of Olivier Robert who passed away while we just cloned the gene. We gratefully acknowledge his dedication to initiate and coordinate this project. We thank Quaid-i-Azam University (Dr. Umar Masood Quraishi, Pakistan), the CIMMYT (Dr. Velu Govidan and Dr. Suchismita Mondal, Mexico), the Dnipro state agrarian and economic university (Dr. Mykola Nazarenko, Ukraine), the National Agricultural Research Institute (Dr. Martín C. Quincke, Uruguay), the National Plant Germplasm System (USA), the Australian Grains Genebank (Australia) and Dr. Richard Oliver, the Wheat Genetics Resource Center (USA), the Biological Resource Centre on Small Grains Cereals (INRAE GDEC, France), KWS (Viktor Korzun) and the Breedwheat consortium for providing seeds of the different wheat accessions. We are grateful to Philippe Lecomte for advices in subcellular localization experiments. We acknowledge the INRA GDEC CPCC, VALFON, and GENTYANE facilities for their technical assistance. We are also grateful to the International Wheat Genome Sequencing Consortium and Kellye Eversole for a pre-publication access to the IWGSC v1.0 wheat genome assembly. We thank Ludovic Bonhomme and Francis Fabre for their help with the Statistical analysis. Our works was supported by the "Fond de Soutien à l'Obtention Vegetale" (FSOV), the French National Institute for Agricultural Research (INRA), and the ETH Zurich Career Seed Grant (SEED-58 16-1). We thank Sreedhar Kilaru and Gero Steinberg for providing the strain 3D7 expressing *eGFP*. Confocal laser scanning microscopy experiments were performed in the Scientific Center for Optical and Electron Microscopy (ScopeM), ETH Zurich and at the SPIBOC imaging facility of the Institut Sophia Agrobiotech. B.F. was supported by INRA and the French Government (National Research Agency, ANR) through the 'Investments for the Future' LabEx SIGNALIFE: program reference #ANR-11-LABX-0028-01.

## Author contributions

C.S., J.F., G.K., O.R., and T.L. conceived the project. F.C. conceived and performed the genotyping, re-sequencing, and complementation experiments. T.P. and F.C. performed *Stb16q* real-time PCR expression experiment. F.C., L.A., E.V., and S.M.T.G. performed phenotyping experiments. W.M. and H.B. developed and screened the BAC library. F.C., M.J., and B.F. did the subcellular localization. S.X. and J.F. developed the first genetic map of *Stb16q*. J.F. generated the EMS-derived mutant population and evaluated the impact of *Stb16q* against SNB, tan spot, and stem rust. J.A. and A.S.V. analyzed the progression of *Z. tritici* on *Stb16q* NILs using confocal microscopy. C.S. drafted the manuscript. J.F., G.K., and T.L. proof-read the manuscript. All authors commented the manuscript.

## Competing interests

The authors declare no competing interests.
