## [Peer Review File · Nature Communications]

REVIEWER COMMENTS

Reviewer #1 (Remarks to the Author):

This manuscript is describing the positional cloning of a novel disease resistance gene, *Stb16q*, conferring broad-spectrum resistance against 64 geographically diverse isolates of the hemibiotrophic and apoplastic fungal pathogen *Zymoseptoria tritici* that is causing *Septoria tritici* blotch (STB) disease of wheat. The authors describe a challenging positional cloning study that was based on an analysis of 9100 F2 plants, comparative genomics, construction and screening of 500,000 BAC clones from the donor line of *Stb16q*. Sequencing of 272 BAC interval revealed only two candidate genes, which one of them, *Crk6*, was validated as *Stb16q* using both loss-of-function (EMS mutagenesis) and gain-of-function (complementation) approaches. *Stb16q* was stated to encode a cysteine-rich receptor-like kinase (CRK) of 684 amino acids with two extracellular copies of the DUF26 domain containing conserved C-X8-C-X2-C motifs, a predicted transmembrane domain and an arginine-aspartate (RD) intracellular protein kinase domain.

This manuscript is describing novel results and I find it suitable for publication in *Nature Communications*, after a minor revision.

My comments are described below:

1. The description of the type of kinase is confusing and needs to be improved. For example in Page 5, line 5 it was stated that *Stb16q* is encoding an arginine-aspartate (RD) intracellular protein kinase domain, while in the legend for Figure 1 (Page 12, Line 1) it is stated to encode "serine/threonine protein kinase, interleukin-1 receptor associated kinase (STKc_IRAK) domains". *Nature Communications* has a wide spectrum of readers and therefore I recommend adding information in the introduction and the discussion that will clarify this issue. For example, I found the following paragraph published in *Genes* by Quezada et al. (2018) as much more clear and much more informative:

"RLKs are categorized into several sub-families, including leucine-rich repeat RLKs (LRR-RLKs), cysteine-rich repeat (CRR) RLKs (CRKs), domain of unknown function 26 RLKs, S-domain RLKs, and others [3]. Cysteine (C)-rich receptor-like kinases (CRKs), also known as DUF26 RLKs, are a large sub-family of plant RLKs. CRKs have a typical RLK domain structure, i.e., they contain an extracellular domain responsible for signal perception, a single-pass transmembrane domain, and a conserved intracellular serine/threonine (Ser/Thr) protein kinase domain responsible for signal transduction. Most CRKs possess two copies of DUF26 in their extracellular domain. The DUF26 domain contains three conserved cysteine residues in a C-X8-C-X2-C configuration."

2. Page 2, Lines 2-3: The last sentence of the abstract is stating that "To our knowledge, this is the first cysteine-rich receptor-like kinase identified as a major resistance gene against plant pathogens."

This information is incorrect since Li et al. (2018) have shown that cotton CRK gene, *GbCRK18*, is conferring resistance to *Verticillium dahlia* (*Front. Plant Sci.* 9:1266. doi: 10.3389/fpls.2018.01266).

Therefore, I would recommend omitting the above statement or limit it to wheat.

3. Page 3, Lines 10-14: "Furthermore, marker development...". This sentence is too complicated and not clear. I suggest simplifying it.

4. Page 3, Line 22: Where all 9000 F2 plants screened with markers for recombination events, or only 1980?

It is not clear, please clarify.

5. Page 4, Lines 6-10: I think this analysis is confusing and it is not adding to the story. I propose

to move it to the end and integrate it with the study of the origin of the gene. Please explain why only the first intron of Crk6 was sequenced.

6. Page 4, Line 25: "by rapid amplification of cDNA ends" – the use of "rapid" is confusing here. I checked and found that it is part of the commercial name of the kit. I propose to change "rapid" to "PCR amplification".

7. Page 5, Lines 1-3: ".....with the avirulent *Z. tritici* isolate IPO09415 and in the susceptible accession ND495..." – this sentence is confusing since isolate IPO09415 is avirulent on resistant accession TA4152-19, but it is virulent on ND495.

8. Page 5, Lines 7-8: "Transient expression assay by agroinfiltration of *Nicotiana benthamiana* leaves of the STB16-GFP fusion protein with the plasma membrane associated AtFH6 protein" – This sentence is confusing. It will benefit the readers to add here that AtFH6 was used as a marker for a membrane associated protein.

9. Page 5, Lines 14-15: "Members of this large multigene family have been identified as candidate genes for resistance against different pathogens" – Members were shown to confer resistance. Please see the comments number 2 above.

Reviewer #2 (Remarks to the Author):

Septoria tritici blotch (STB) is an important foliar disease in wheat and is also one of the difficult ones to control. The authors used map-based cloning and cloned a major resistant gene *Stb16q*, and used comparative genomics, mutagenesis and transformation to confirm. *Stb16q* confers broad-spectrum resistance and no fully virulent pathogen has been identified. Therefore, wisely deploy this gene will help protect wheat crops from STB. It also has a unique structure of being a cysteine-rich receptor-like kinase, which is different from all the cloned plant resistance genes for far. Its cloning will help in understanding the gene functions and molecular mechanisms involved in resistance. This paper is well-written and experiments are done beautifully. It provides new information from both theoretical and applied point of view. I suggest that it can be published with minor revisions.

The format of the manuscript that it is currently written is a bit different from the papers published in Nature Communications. I suggest that they rearrange different sections in the order of Introduction, Results, Discussion, and Methods, rather than not list Methods in the paper itself and has its own set of references. Combining them into one complete paper will flow better.

P3, Para2, L13: scaffold45305 is not in Suppl. Fig. 2.

P3, Para2, L16-17 & figure legend for Fig. 1: From Fig. 1, the genetic distance between the two markers are 0.02 cM instead of 0.07 cM.

When the number is bigger than 10, it does not need to be spelt out.

The authors only identified two susceptible M2 plants from the same family, which basically came from the same mutational event. Normally, more than one mutant is needed in order to confirm its function. In this case, susceptible mutant has the change in the gene candidate. Furthermore, they have clear stable transformation results and other evidence, which they may get away with only

one mutation.

P4, Para1, L21: It should be "avirulent Z tritici isolates", not the "virulent" ones. The same applies to Fig. 1, which should be avirulent isolates.

The first letter in "Spring" as in "Chinese Spring" needs to be capitalised.

P20, L1-2: The expression of "was increased twice by 1X" is a bit confusing and needs rewording.

In the figure legends for Suppl. Fig. 6: It should be "Tyr-Val" instead of "Tyr-Via".

REVIEWER COMMENTS

Reviewer #1 (Remarks to the Author):

This manuscript is describing the positional cloning of a novel disease resistance gene, Stb16q, conferring broad-spectrum resistance against 64 geographically diverse isolates of the hemibiotrophic and apoplastic fungal pathogen *Zymoseptoria tritici* that is causing Septoria tritici blotch (STB) disease of wheat. The authors describe a challenging positional cloning study that was based on an analysis of 9100 F2 plants, comparative genomics, construction and screening of 500,000 BAC clones from the donor line of Stb16q. Sequencing of 272 BAC interval revealed only two candidate genes, which one of them, Crk6, was validated as Stb16q using both loss-of-function (EMS mutagenesis) and gain-of-function (complementation) approaches. Stb16q was stated to encode a cysteine-rich receptor-like kinase (CRK) of 684 amino acids with two extracellular copies of the DUF26 domain containing conserved C-X8-C-X2-C motifs, a predicted transmembrane domain and an arginine-aspartate (RD) intracellular protein kinase domain.

This manuscript is describing novel results and I find it suitable for publication in Nature Communications, after a minor revision.

My comments are described below:

1. The description of the type of kinase is confusing and needs to be improved. For example in Page 5, line 5 it was stated that Stb16q is encoding an arginine-aspartate (RD) intracellular protein kinase domain, while in the legend for Figure 1 (Page 12, Line 1) it is stated to encode “serine/threonine protein kinase, interleukin-1 receptor associated kinase (STKc_IRAK) domains”.

Nature Communications has a wide spectrum of readers and therefore I recommend adding information in the introduction and the discussion that will clarify this issue. For example, I found the following paragraph published in Genes by Quezada et al. (2018) as much more clear and much more informative:

“RLKs are categorized into several sub-families, including leucine-rich repeat RLKs (LRR-RLKs), cysteine-rich repeat (CRR) RLKs (CRKs), domain of unknown function 26 RLKs, S-domain RLKs, and others [3]. Cysteine (C)-rich receptor-like kinases (CRKs), also known as DUF26 RLKs, are a large sub-family of plant RLKs. CRKs have a typical RLK domain structure, i.e., they contain an extracellular domain responsible for signal perception, a single-pass transmembrane domain, and a conserved intracellular serine/threonine (Ser/Thr) protein kinase domain responsible for signal transduction. Most

CRKs possess two copies of DUF26 in their extracellular domain. The DUF26 domain contains three conserved cysteine residues in a C-X8-C-X2-C configuration.”

Response: Thank you for pointing this out. We changed the description of STB16 (P5) as “an arginine-aspartate (RD) intracellular serine/threonine (Ser/Thr) protein kinase domain” and changed as well the text in the legend of Figure 1. Furthermore, we added a detailed description of this gene family in the discussion.

2. Page 2, Lines 2-3: The last sentence of the abstract is stating that “To our knowledge, this is the first cysteine-rich receptor-like kinase identified as a major resistance gene against plant pathogens.”

This information is incorrect since Li et al. (2018) have shown that cotton CRK gene, *GbCRK18*, is conferring resistance to *Verticillium dahliae* (Front. Plant Sci. 9:1266. doi: 10.3389/fpls.2018.01266).

Therefore, I would recommend omitting the above statement or limit it to wheat.

Response: A major resistance gene is defined as a gene, which confers qualitative resistance. The cotton CRK *GbCRK18* was shown to play a role in resistance to *Verticillium dahliae* as silencing this gene by virus-induced gene silencing mediates plant susceptibility. However, it is not known if this gene would confer resistance once introduced into a susceptible accession, as Li et al. did not transfer this gene under its natural promoter into a susceptible accession. Therefore, we cannot say that *GbCRK18* confers resistance but we can say that it is involved in resistance. In our study we showed that *Stb16q* confers resistance to *Z. tritici*. To avoid any confusion, we modified the last sentence of the abstract to “To our knowledge, this is the first cysteine-rich receptor-like kinase cloned, which confers major resistance against plant pathogens.” Furthermore, we added a small paragraph in the discussion to explain the statement described above regarding *GbCRK18*.

3. Page 3, Lines 10-14: “Furthermore, marker development...”. This sentence is too complicated and not clear. I suggest simplifying it.

Response: Done.

4. Page 3, Line 22: Where all 9000 F₂ plants screened with markers for recombination events, or only 1980?

It is not clear, please clarify.

Response: Only 1,980 F₂ plants (the most resistant and the most susceptible) were screened with markers for recombination events as it is mentioned in the text “New markers designed from both scaffolds were used to genotype 1,305 immune and 675 highly susceptible F₂ plants selected from a large-scale phenotyping experiment of 9,100 F₂ plants”. To clarify, we replaced “genotype” by “search for recombination events among”.

5. Page 4, Lines 6-10: I think this analysis is confusing and it is not adding to the story. I propose to move it to the end and integrate it with the study of the origin of the gene. Please explain why only the first intron of *Crk6* was sequenced.

Response: This paragraph represents the study of the association between SHWs phenotyping data and sequencing data of both candidate genes from these SHWs. It is right that this analysis did not help us to discriminate between both candidates. We decided to keep this paragraph as it allows presenting the genetic diversity of the Unk gene. However if you think that it is still not necessary we can remove it. All sequencing data present in this paragraph is included in the study of origin. Only the first exon of Crk6 was sequenced as it represents its most genetically diverse region as illustrated by sequencing data of the parents and the literature on RLK genetic diversity. We added this explanation in the paragraph.

6. Page 4, Line 25: “by rapid amplification of cDNA ends” – the use of “rapid” is confusing here. I checked and found that it is part of the commercial name of the kit. I propose to change “rapid” to “PCR amplification”.

Response: Done.

7. Page 5, Lines 1-3: “.....with the avirulent *Z. tritici* isolate IPO09415 and in the susceptible accession ND495...” – this sentence is confusing since isolate IPO09415 is avirulent on resistant accession TA4152-19, but it is virulent on ND495.

Response: Right, we modified it by “the *Stb16q* avirulent *Z. tritici* isolate IPO09415”.

8. Page 5, Lines 7-8: “Transient expression assay by agroinfiltration of *Nicotiana benthamiana* leaves of the STB16-GFP fusion protein with the plasma membrane associated AtFH6 protein” – This sentence is confusing. It will benefit the readers to add here that AtFH6 was used as a marker for a membrane associated protein.

Response: We modified the sentence by “Transient expression assay by agroinfiltration of *Nicotiana benthamiana* leaves showed colocalisation of the STB16-GFP fusion protein with AtFH6-mCherry, a plasma membrane-associated protein used as a marker²⁶, confirming its predicted localization at the plasma membrane (Supplementary Figure 7)”.

9. Page 5, Lines 14-15: “Members of this large multigene family have been identified as candidate genes for resistance against different pathogens” – Members were shown to confer resistance. Please see the comments number 2 above.

Response: As mentioned in response to comment 2, we added a paragraph in the discussion stating that *GbCRK18* is involved in resistance but we do not know if it will confer resistance once transferred into a susceptible accession.

Reviewer #2 (Remarks to the Author):

Septoria tritici blotch (STB) is an important foliar disease in wheat and is also one of the difficult ones to control. The authors used map-based cloning and cloned a major resistant gene *Stb16q*, and used comparative genomics, mutagenesis and transformation to confirm. *Stb16q* confers broad-spectrum resistance and no fully virulent pathogen has been identified. Therefore, wisely deploy this gene will help protect wheat crops from STB. It also has a unique structure of being a cysteine-rich receptor-like kinase, which is different from all the cloned plant resistance genes for far. Its cloning will help in understanding the gene functions

and molecular mechanisms involved in resistance. This paper is well-written and experiments are done beautifully. It provides new information from both theoretical and applied point of view. I suggest that it can be published with minor revisions.

The format of the manuscript that it is currently written is a bit different from the papers published in Nature Communications. I suggest that they rearrange different sections in the order of Introduction, Results, Discussion, and Methods, rather than not list Methods in the paper itself and has its own set of references. Combining them into one complete paper will flow better.

Response: Done

P3, Para2, L13: scaffold45305 is not in Suppl. Fig. 2.

Response: Scaffold45305 is not in Supplementary Figure 2 as it does not carry any of the 42 SNP markers identified between *barc323* and *cf9*. It is however present in the physical interval defined by Jia et al. We added a sentence in the legend stating, “Among these latter, scaffold45305 is present between scaffold1066 and scaffold39406”.

P3, Para2, L16-17 & figure legend for Fig. 1: From Fig. 1, the genetic distance between the two markers are 0.02 cM instead of 0.07 cM.

Response: From Figure 1, the genetic distances between marker *cf335* and *Stb16q* is 0.02 cM and 0.05 cM between *Stb16q* and marker *cf80033*. Therefore, the genetic distance between *Stb16q*-flanking markers *cf335* and *cf80033* is 0.07 cM.

When the number is bigger than 10, it does not need to be spelt out.

Response: ok, done. We modified it (P4, P5 and P6).

The authors only identified two susceptible M₂ plants from the same family, which basically came from the same mutational event. Normally, more than one mutant is needed in order to confirm its function. In this case, susceptible mutant has the change in the gene candidate. Furthermore, they have clear stable transformation results and other evidence, which they may get away with only one mutation.

Response: This is right. We have only two M₂ plants from the same family identified as susceptible which share the same mutational event. For sure, one mutational event is not enough evidence to say that *Crk6* is a candidate. However, we analyzed 54 progenies from one of the M₂ plants (heterozygous for the mutation) and showed that homozygosity of this mutation co-segregates with plant susceptibility. This analysis strengthens that this mutational event maybe responsible of the susceptible phenotype even if we cannot rule out the impact of a closely linked mutation. We think it is worth keeping this paragraph.

P4, Para1, L21: It should be “avirulent Z tritici isolates”, not the “virulent” ones. The same applies to Fig. 1, which should be avirulent isolates.

Response: It is indeed confusing. P4 L21: The term virulent was used to state that these isolates were virulent against a large number of *Stb* genes. To avoid any ambiguity, we

removed the terms virulent or avirulent.

The first letter in “Spring” as in “Chinese Spring” needs to be capitalised.

Response: Done. Modified in P6 and P12, Legends of Supplementary Figure 1 and Supplementary Figure 4 and Materiel and Methods.

P20, L1-2: The expression of “was increased twice by 1X” is a bit confusing and needs rewording.

Response: Done, we made two sentences. “To fill in the physical gap, depth of coverage of the Tae-B-4152-ng BAC library was increased twice. Each time, a one-fold coverage of the genome was achieved.”

In the figure legends for Suppl. Fig. 6: It should be “Tyr-Val” instead of “Tyr-Via”.

Response: Done.

REVIEWERS' COMMENTS

Reviewer #2 (Remarks to the Author):

The authors have responded to both reviewers' comments satisfactorily. They have made corresponding changes and improvement on the manuscript, which is much clearer on several points raised by the reviewers. I think the manuscript can be accepted.